# Sound Feedback for Social Distance: The Case for Public Interventions during a Pandemic

**William Primett** [1,*] **, Hugo Plácido Da Silva** [2] **and Hugo Gamboa** [1]

1   LIBPhys Faculdade de Ciências e Tecnologia, Universidade Nova de Lisboa, 2829-516 Caparica, Portugal; hgamboa@fct.unl.pt
2   Instituto de Telecomunicações (IT), 1049-001 Lisbon, Portugal; hsilva@lx.it.pt
*   Correspondence: w.primett@campus.fct.unl.pt

**Abstract:** Within the field of movement sensing and sound interaction research, multi-user systems have gradually gained interest as a means to facilitate an expressive non-verbal dialogue. When tied with studies grounded in psychology and choreographic theory, we consider the qualities of interaction that foster an elevated sense of social connectedness, non-contingent to occupying one's personal space. Upon reflection of the newly adopted social distancing concept, we orchestrate a technological intervention, starting with interpersonal distance and sound at the core of interaction. Materialised as a set of sensory face-masks, a novel wearable system was developed and tested in the context of a live public performance from which we obtain the user's individual perspectives and correlate this with patterns identified in the recorded data. We identify and discuss traits of the user's behaviour that were accredited to the system's influence and construct four fundamental design considerations for physically distanced sound interaction. The study concludes with essential technical reflections, accompanied by an adaptation for a pervasive sensory intervention that is finally deployed in an open public space.

**Keywords:** wearable sensors; interpersonal movement; pervasive technology; social computing; public space

## 1. Introduction

The Coronavirus Disease 2019 (COVID-19) was affirmed as a pandemic by the World Health Organization (WHO) on 11 March 2020 [1] presenting numerous unforeseen novelties given that symptoms would range from severe to unrecognisable over an indeterminate timeline [2]. As the effects of the pandemic intensified on a global scale, central governments enforced new regulations that compromise public space interactions against the preceding normality. While the specific constraints varied between countries and eased accordingly over time, a key component throughout was to prohibit physical contact almost entirely where viable and to avoid close proximity with others [3,4]. Simultaneously, the sudden ubiquity of protective face coverings has introduced a plethora of unanticipated complications regarding verbal and non-verbal expressions that are habitually relied upon to interpret emotional states [5–7].

In accordance with these measures, a vast majority of public events were jeopardised, particularly in the case of live performances and artistic installations. As pandemic regulations were gradually alleviated, however, public activities were re-authorised on the grounds that all participants complied with mask-wearing and distancing orders [8]. We foresaw a need to address the tensions surrounding safe social conduct as well as the risks of viral transmission newly associated with on-body sensor technologies [9,10]. The physical space observed between bodies can be linked to various social cues during an interaction [11–13]. However, these inferences can deviate massively between individuals and therefore it is unintuitive to commit to a one-size-fits-all model that is wholly representative of all sorts of social contexts and cultural environments [14,15].

We propose an intervention to challenge the conventional dualisms between physical distancing and social connectedness, based on sound and movement. This begins by reconstructing our approach to both sensor-based monitoring and the delivery of live performance, looking beyond the introspective experience, and considering affective processes that occur externally from the body. We then present a wearable system for sound–movement interaction centralised around interpersonal distance and collective movement. Following a series of ideation sessions, over a two-week period, the system was situated in a live performance setting from which we evaluate the performer's quality of movement in relation to the sound feedback. We describe some of the system's key influences contextualised in the socio-emotional domain, inducing entrainment, spontaneity and awareness. We formulate the following research questions: RQ1 How can sound feedback improve collective awareness and sensibility to interpersonal distance? RQ2 What movement patterns are influenced by external mediation materials? and RQ3 What technological interventions are appropriate for proxemic interaction in public space?

Our work continues to extend the rich design space for collective sound environments with movement sensing. We outline our perspectives on mediating social interactions with regard to the proposed coupling of distance sensing with sound actuation. Following this, we acknowledge the vital limitations of our initial framework and put forward research opportunities that may arise from extended exploration. Namely, the incentive of transitioning into open public environments where participation can occur pervasively. Additionally, we document the necessary measures taken with respect to the restrictions put into force by the COVID-19 pandemic, the impact of which remains prevalent up until the time of writing. Howbeit, our findings should be interpreted as universally relevant toward orchestrating spatially aware interventions even in non-pandemic circumstances.

## 2. Background

### 2.1. Social Distancing and Public Presence

It is evident that regular social engagement plays an important role in community wellbeing. Not only bonding with our close companions in organised situations but inclusive of unanticipated interactions that emerge in public [16–18]. The first wave of the COVID-19 pandemic elevated some attention to this. To exist outside of our own home meant weighing out the chances of a life-threatening infection against the consequences of avoiding social connection, endangering one's health in other ways [19–22].

As emergency COVID-19 measures were becoming standardised globally, epidemiologists from the WHO firmly recommended a linguistic shift from 'social' to 'physical' distancing on the basis that modern technology is capable of keeping us connected in spite of the new regulations [23]. Given that the anticipation of the first wave would in essence reject in-person contact in the way it was known; this narrative was therefore appointed to the normalisation of fully remote communication. However, as society would phase in and out of non-confinement periods, conventional face-to-face interaction could take place again, persisting with the caution of physical distance, embracing *Social Proximity* as a safe practice for social wellbeing [24]. Irrespective of this major adaptation to safer re-socialisation, we argue that the development of in-person interventions has not matured to the same extent as remote interaction technologies and that there still lies a vast design space yet to be fulfilled.

### 2.2. Proxemic Behaviour and Re-Socialisation

The attention toward interpersonal distance coincides with the study of proxemics, examining the function of physical space during face-to-face interactions [25,26]. Proxemic theory has been given a lot of attention in a wide range of behavioural studies [27] and thus spurs incentives from several ubiquitous computing projects [28,29]. A majority of these accept the proxemic zones set out by Hall [25], by which interpersonal distances are generally categorised into the following boundaries: Intimate, up to 1.5 feet (0.45 m); Personal, 1.5 to 4 feet (1.2 m); Social, 4 to 12 feet (3.6 m); and Public, more than 12 feet

(7.6 m). We believe, however, that the rich contextual nature of interpersonal movement behaviour poses a challenge for conventional computational modelling, calling for expertise to break down and articulate expressive features of human motion and sensation [30,31]. With regard to proxemics, we cannot solely depend upon the measure of distances and angles to define the affective characteristics that occur during any given exchange.

Coping with the current risks of infection in everyday social contact, the new *sociable space* implies a new urban etiquette that expands upon the standardised dimensions proposed by Hall's proxemic theory, preserving conversational affordances at double the distance [32]. In a general sense, spontaneous, informal encounters are recognised as an essential part of growing one's social circle, fostering a sense of belongingness within the community [33]. However, the circumstances in which these are likely to occur are vulnerable to the spatial dynamics and physical distance between bodies [34,35]. These types of relationships fit into the social formation of indirect contact, which can take place seamlessly by way of intergroup behaviour, alleviating the pressures of face-to-face enactment [36] that are increasingly present in the midst of pandemic concerns [37]. While, of course, many will long to reconnect with their peers during moments of close physical bonding, these new sociable spaces provide an enlightening deconstruction of proxemic acceptability where strangers and non-strangers are both welcomed into everyday social encounters. The linguistic connotation of 'distancing' assumes active separation from others, while its function should instead be geared towards social affirmation, organised in a safe manner. We propose an interactional view on social distancing that is unconditional to definitive measurements, advancing upon the outlooks contained in the following study [38] that insists on expanding Hall's discrete interpretation of proxemic zones when applied to continuous mappings of movement.

The orchestration presented in this case study is set out to capture the affective outcomes when proxemic behaviours are exaggerated during social exchange, where sensory intervention goes beyond being an assessment tool, but becomes a modality for non-verbal expression.

### 2.3. Sensor Technology and the Right to the City

When considering proxemic behaviours in public, urban pedestrian areas have long exemplified the effectiveness of street-installed technology to manage the mobility of crowds; however, the vital function is vulnerable to being overthrown in the case of high-intensity overcrowding [39,40], which has been associated with feelings of anxiety, frustration and claustrophobia [41]. Certain demographics are more sensitive, while others are far less cautious of invading the personal space of others [42,43]. With respect to interpersonal distance, initiatives in response to pandemic conditions rely upon a level of altruistic responsibility, by which the public are inclined to follow a protective etiquette [44]. When taking a look at the technologies that have become ubiquitous in response to the pandemic: Contact Tracing Applications, Skin Temperature Scanners, and contactless patient monitoring, for example [45,46], we note that the user is constrained to one interpretation of the data made available to them. When contrasted with alternative interventions that explore emotional bonding in urban environments, e.g., [47,48], we begin to question the limitations of the prior, and subsequently, practices that exist at the intersection of social affirmation and long-lasting public health. Building upon the insights presented by Howell [49], calling for a progressive turn in public space sensing technologies [49], we insist that the data-driven motives pushed onto current smart city development schemes are non-compliant with these sorts of shared emotional experiences [50–52].

The long-standing notion known as "the right to the city" has been embraced by Interaction Design researchers, commentating on the industrialised approach that we are seeing with sensor technology that is being increasingly embedded into urban spaces, shifting the perspective from data-driven smart cities to 'social' or 'playful cities' [49,53]. Such case studies express a necessity for inclusive participation to establish grounds for progressive social integration, framed as the collective right to be in control of the surroundings through

co-creation. Acting upon the current situation, we enquire into a sensory intervention that aids awareness of one's physical presence and self-affirmed boundaries, without the need to declare any action as right or wrong.

### 2.4. Appropriation of Social Distance and Interpersonal Touch

The urgent nature of the pandemic demanded disruption to common interactional norms, namely the concern of keeping distance and avoiding touch [24,54]. However, despite a reactionary appeal for spatially-aware behaviours, this call for action has been relevant for a while now. There exist several rationals for designing systems that are considerate of one's personal space, which in our view, have been neglected by relevant research areas.

For many, the apprehension of close contact in urban space serves as an evaluation criterion for everyday safety [42,55]. A preference for greater interpersonal distances is also evident for those suffering from anxiety disorders, commonly assumed to an avoidance of social interactions entirely [56]. Intersecting this view with the subjective quality of skin-to-skin contact, praised by a substantial volume of research for inciting profound therapeutic sensations [57], presented as an appealing method to relieve anxiety, strengthen social bonds, and even elicit physical health benefits [58–60]. However, this is not always the case if we take into account, for instance, those who experience hypersensitivity (or lack of) to introspective stimuli [61,62]. Furthermore, the perceived benefits associated with affective touch are largely subject to the qualities of the pre-existing relationship of the dyad [63].

Drawing parallels between inclusive proxemic interventions and safe re-socialisation [24], we want to understand the liberties that can emerge from contact-less mediation while protecting the right to physical presence. Therefore, it is important to depart from the assumption that everybody is entirely comfortable with close contact, and understand that the degree of comfort largely depends upon who is approaching them and in what context [64,65]. In Section 5, we discuss the importance of flexible boundaries when designing for interpersonal distance, conditional to the social context at hand and the proxemic sensitives of the individual.

### 2.5. Non-Verbal Contingencies and Face Coverings

Proxemic interactions, given the alliance with mutual gaze, would assume the inclusion of facial expressions, where conventionally, more salient emotional attributes are depicted between the nose and the chin [66,67]. The visibility of facial expressions and clarity of speech is both highly relied upon in everyday communication, but the regulated use of the face mask disregards this modality almost entirely [6]. This becomes even more crucial in the context of physical distancing regimes [68], where proxemic studies report a habitual reduction of eye contact when spaced more than twelve feet (3.6 m) apart, i.e., public distance [69]. Additionally, studies related to pandemic behaviour have demonstrated how face masks influence interpersonal distances [70]. Our facial muscles can expose a great deal of how we are feeling [71,72], sometimes completely unknowingly to the extent that one may exert themselves into forcibly concealing these expressions while under pressure [73]. In comparison, how we move in space is normally a consequence of deliberate coordination during an interaction [74,75]. Hadley and Ward [76] point close attention to the physical gestures used as a substitute for verbal exchange where background noises impair vocal comprehension [76]. The normalisation of face masks worn during conversation has been shown to degrade the acoustic quality of the voice as a result of suppressing higher frequency ranges, commonly depended upon to recognise articulations of consonant sounds [77–79]. This poses a further disadvantage to those hard of hearing [80] as well as non-native speakers, often more dependent on reading the face [81,82].

In Section 3.3, we outline our fabrication methods, taking the newly ubiquitous face mask, commonly condemned as a social hindrance and reshaping this as sensory material for non-verbal exchange, isolating communication channels aside from the face.

## 2.6. Measuring Interpersonal Movement

Aside from interpersonal distance, we are also interested in movement qualities that can characterise aspects of a social situation. Such qualities are recognised by Rudolf Laban's Movement Analysis (LMA), a well-established notation framework used to depict expressive features of human movement, originating from the perspective of dance and physical therapy [83], and since adapted to all kinds of contexts (e.g., routines of factory workers) [84]. With ongoing advancements toward body-centred applications, Laban's theory on movement has firmly settled itself into Human–Computer Interaction research [85].

It is apparent here that the intersections of HCI and Laban Movement principles tend to focus on individual accounts. Pluralist qualities, on the other hand, comprised of two or more persons moving together, are observed as the *Relationship* category. The following article from Roudposhti et al. affirms a scarcity of literature in this domain [86], expressing the unexplored potential for social interactive systems. The authors propose a global feature space that combines Pentland's analysis model of social signals [87] with LMA qualities, taking upon the following descriptors: *Indicator*, *Empathy*, *Interest* and *Emphasis*. In Section 5, we borrow two of these qualities in our evaluation, *Indicator* to describe the exchange between influent and influenced members, presumed by the difference in energy between them, and *Interest* representing one's engagement to the situation or outside context, gauged by energetic movements. Similarly to Laban's Movement Analysis, each quality operates on a continuous scale between two polarities.

## 2.7. Sound Interaction as Social Mediation

Throughout the extensive literature surrounding proxemic interfaces, we came across a surprising lack of studies related to sound, given this is already an established modality for movement interaction with ties to affective representation [88], supported by a base understanding that physical action and sound perception are mutually responsive [89]. As a tentative presumption, we can point to the inherent limitations of distance detection with typical camera-based tracking technologies [90] as well as the obtrusiveness of on-body sensor devices [91]. An affirming study that fits into this criteria installs a proxemic augmentation into a gallery space [92], supporting the role of proxemic audio interactions in a "post-screen world" [93]. A recent study comparing common actuation modalities by Alfaras et al. demonstrates the usefulness of audio-based biofeedback to foster physiological synchronisation and somatic awareness, noting that the human hearing system is highly sensitive and that sound is a convenient medium to share amongst many users [94]. Furthermore, recent literature suggests that synchronous motor activity can be indicative of prosocial affiliation [76], particularly when contextualised with sound, be it disruptive or complimentary [95].

Within the context of music performance, the concept of a collaborative system is not a novel phenomenon, far from it in fact. Reports date back as far as 1978 [96] with progressions to real-time remote interaction [97], later supporting the establishment and maturity of interactive music systems (IMSs), e.g., [98,99]. A recent review from Aly et al. discusses the pervasive nature of sound interactions in the context of biosignal-driven IMSs when capturing data from many users simultaneously [100]. Giving attention to collective movement, Hege presents an artefact by which the members of the Princeton Laptop Orchestra showcase democratic expertise through intentional yet delicate control of the sound output [101]. IRCAM researchers advocate for a human-centred framework for gestural sound control in a group scenario [102], not only for performance but also justified in clinical use-cases [103].

Moving away from the audience–performer dynamic, related case studies demonstrate how social encounters can be mediated through embodied sensor data [49,104,105]. We would like to further investigate this approach to engage multiple users simultaneously, in this instance, representing an assembly of interpersonal distances through sound.

### 3. Materials and Methods

#### 3.1. Composition of a Wearable Sensing Medium

We decided to design a wearable sensing medium centred around the face mask. These were a mandatory possession for local citizens that had already become a cultural norm for one's public appearance [106]. Our pursuit towards an on-body device was supported by Montanari et al.'s investigation into a novel proxemic sensor, welcoming a sacrifice in the high-level attributes that come with camera-based tracking in preference of environmental flexibility, quoting the affordance "*to collect data even in areas that cannot be instrumented, like public spaces or during large events*" [107]. We consider the worthwhile benefits of a wearable solution that is more versatile in non-laboratory situations [95,108,109]. In essence, this bypasses the challenges articulated by Jürgens et al. directed towards using an unobtrusive markerless motion capture technology in an on-stage environment for contemporary dance performance [110]. The authors highlight digital errors inflicted by particular lighting conditions, clothing contrast, as well as scenarios where performers were in close proximity to each other. For our study, we were incentivised to capture the point of view of the user, aligning the sensing trajectory with the user's gaze during an interaction, as detailed in the following sections.

#### 3.2. Overview of Components

We have interfaced low-cost HC-SR04 ultrasonic distance sensors with the BITalino R-IoT microcontroller using a modified firmware (included in Appendix A); this allows the acquisition of proxemic data from multiple participants at 10 samples per second (s). When initiated, each module streams data wirelessly over a designated local Wi-Fi network to a host computer, which is then responsible for signal processing and sonification. With this sampling rate, the maximum response time is accepted as 100 milliseconds (ms) plus any wireless latency, averaging at 10–11 ms in such conditions (report provided in Appendix B). This sensing technology is highly prevalent in robotics and IoT educational fields, typically used as part of introductory curricula [111,112] while sustaining relevance in the state of the art (e.g., [113,114]). Fundamentally, the sensor measures the distance from the first physical interference at a given direction by transmitting and receiving ultrasonic frequencies outside of the human hearing range [115]. The range and accuracy benchmarks for the ultrasonic sensors are partially dependent on environmental variables such as ambient temperature and humidity. When working in typical indoor conditions, the ultrasonic sensors are expected to ensure a stable accuracy of up to 13 feet (4 m), within a 15 degree angle [116].

With respect to providing a wearable form factor, and minimising the size and weight of the components, we implemented a discrete voltage divider onto the sensor's Ground and Echo pins to comply with the power specifications commonly found in smaller microcontrollers, usually rated at 3.3 Volt. With this configuration, there was a minor but notable drop in accuracy compared to our tests using 5-Volt compatible microcontrollers. However, this was mostly resolved with signal processing to the extent that the effects would not hinder the overall sound-based mediation experience. Following technical directions from Kielas-Jensen [117], we were able to incorporate data from the R-IoTs embedded temperature sensor into the distance acquisition function, this improved the reliability and consistency of the readings when transitioning between distinct environmental conditions. For example, a computer lab, open exhibition space, performance theatre holding maximum occupancy, or even installed in an open outdoor area.

#### 3.3. Material Design and Fabrication

In order to publicly distribute the sensory masks in a safe manner, respecting the government guidelines, we assigned the following design principles: (1) The sensory components are modular and detachable, allowing for sanitation while the mask fabric is being replaced; (2) These components must remain at a fixed position and be robust in situations of rapid movement; and (3) The mask should feel sufficiently comfortable for users to

wear for prolonged periods of time. Additionally, for the system to accommodate mass participation, we favoured a scalable solution that was low-cost and easily reproducible (4). For each wearable, we modelled two separate housing elements for the respective components, i.e., the microcontroller and the proximity sensor. The 3D models are included in Appendix A.

For the sensing mechanism, the ultrasonic sensor casing was merged with an arched nose clip and secured onto the mask fabric. The microcontroller and battery were encased in a custom mask strap that would be tied around the back of the head. The components are connected via a flexible cable, splitting into four wires each connecting to the corresponding inputs of the sensor and microcontroller, transmitting analogue signals back-and-forth to retrieve proximity data, represented as trigger and echo in Figure 1. These are assembled by positioning the sensor component above the nasal dorsal, resting front-most of the face, in line with the gaze with minimal obstruction. To overcome the precariousness of the sensors dropping down from the nose during movement, we increased the tension of the elastic ear straps and fastened a flexible rod between the mask fabric and sensor attachment. While this modification improved the structural reliability, our capacity to freely substitute sensory modules, along with spontaneous user alterations was limited as a consequence.

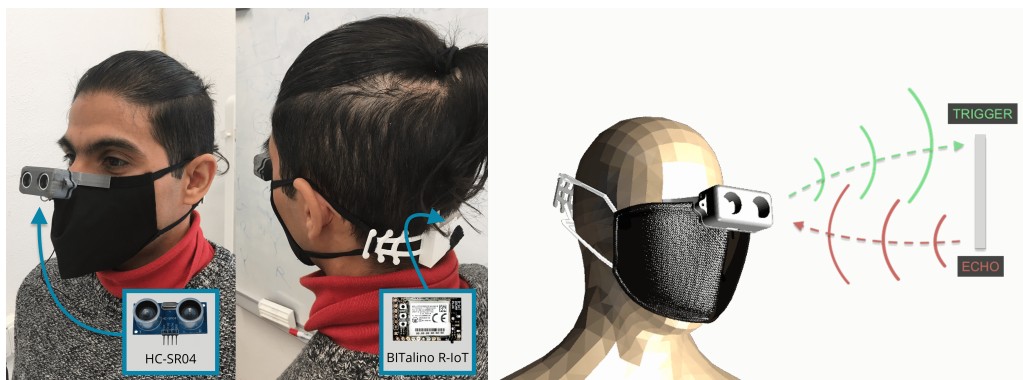

**Figure 1.** Proximity sensor enclosure fitted onto the face mask with trigger (output) and echo (receiver) signals.

### 3.4. Data Processing and Sound Mapping Strategies

The orchestration does not require participants to conform to a fixed minimum distance. We lend our trust to the users to coordinate themselves in a safe manner without explicit orders, as it is expected in their daily life. Prompted by RQ1, the sound interaction is purposed to strengthen one's awareness of the other's presence, unbounded from discrete categorisation, engaging one's auditory senses while verbal modalities are restricted. In addition to this, the experience should not call for virtuosity or a specialist training process. The system assumes engagement from the moment the user is being sensed and become progressively accustomed to the sound–movement affordances. This aligns with the presumption that the participants are expected to work together to develop their understanding of the system and stay vigilant to each other's actions.

During these first trials, we used granular synthesis (Grain Scanner by Amazing Noises: https://www.ableton.com/en/packs/grreciprocatein-scanner-amazing-noises/ (acessed on 6 July 2022) to interpolate between 4 field recordings as base sound textures. Inspired by Laban's qualities of space and relationship, we proposed a transition between "open" to "closed" sound textures, continuously modulated in real-time as the average distance reaches the minimum sensing boundaries of 4 feet (1.2 m), before succeeding social distance.

This process starts with a lot of fluctuations in timbre and then closes in on a limited range of frequencies that form into a staccato rhythmic feel. A one minute sound sample is visualised in Figure 2.

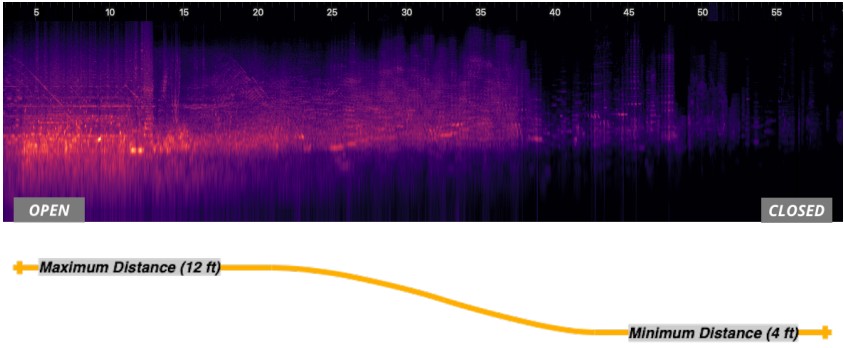

**Figure 2.** Spectrogram sample of interactive soundscape showing "open" to "closed" transition.

*3.5. Performance Structure*

The system's initial deployment exposed a significant level of confusion from new users when given the chance to freely experiment with the device. Aside from the technical ideation, we speculated upon ways of stimulating movement whilst sustaining an appropriate degree of improvisational freedom. This reiterates that we were not interested in forcing a strict sequence of movement, but rather providing geometrically informed cues set out to prompt dyadic gestures. We devised three scenes that each focus on distinct affordances of proxemic mediation, as illustrated in Figure 3 and presented in the following order:

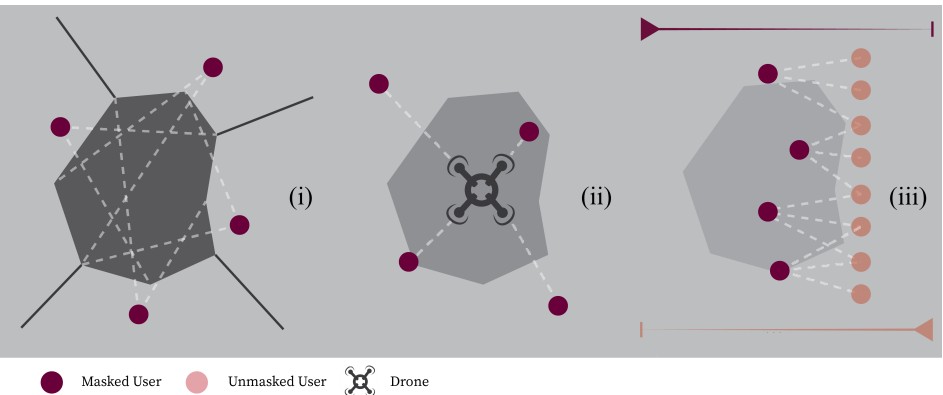

**Figure 3.** Visual representations for each scene. From left to right: (i) Geometric boundaries and interceptions, (ii) Interacting with the non-human, (iii) Participation from external users.

The group was made aware of the performance structure and interactive elements described in Section 3.2, while granted the freedom to explore the space and move intuitively to the sound feedback, whilst maintaining a minimum interpersonal distance of 5 feet (1.5 m). For these actions, a 225 m² performance area was used accompanied by two concert-grade loudspeakers.

3.5.1. Performers and Objects

These geometric arrangements are predominantly inspired by William Forsythe's architectural approach to choreographic environments [118], a radical staple in contemporary dance culture, bridging deeply into other artistic mediums [119].

Giving attention to RQ2, the following passage redirects the proxemic attention from the neighbouring bodies, onto the surroundings (Figure 3(ii)), supported by Kinns's recommendations for mutual engagement with shared representations [120]. In the extended proxemic criteria presented by [38], the relationship to other objects, both digital and non-digital, is embraced. Adopting theories of gaze-based interaction into the format of proxemic awareness, authors claim to enrich the user's attention to the space by attending to implicit responses from the user's surroundings.

Autonomous flying drones have been incorporated into a stage performance [121] and movement-centred practices [122] proceeding efforts to enhance kinesthetic awareness through intercorporeal engagement [123]. During Scene ii, the drone represents more generally an unpredictable external influence, comparable to the external nuances that occur in public space environments.

In relation to this, Ballendat's evaluation of screen-based proxemic interactions considers the attention directed to other people, allowing for conventional social exchange to coexist alongside the technological artefact itself, accepting natural influences that would arise in coherence with an everyday social situation [38].

### Scene I, Geometric Boundaries and Interceptions

To begin, the performers position themselves around the boundaries of the performance area, facing the centre (Figure 3(i)), standing 18 feet (5 m) apart. The first sounds are activated as users cross to the opposite side, alternating back and forth until coinciding with the linear path of one another, after which an anticipated diversion is required. One participant offers the following interpretation: "*The soundscape intensifies as the collective tightens in space. Before collision, we have to figure out our next steps, to dodge, retrace or simply pause for a moment*". Our improvisational framework here is designed to inspire proxemic exchange, while the primary responsibility of avoiding near-contact is handed over to the performers.

### Scene II, Interacting with the Non-Human

A technician is assigned the role of navigating the stage with a flying drone, weaving between the masked performers. When advancing towards the bodies on stage, specifically targeting the sensory components, the drone serves as an external trigger as it reaches close enough to the face. As a result, this synthetic artefact would provoke instantaneous reflex responses from the performers that consequentially would instigate dynamic changes in the soundscape.

### Scene III, Participation from External Users

Finally, we invite external users that are not individually equipped with any wearable sensors but are authorised to manipulate the behaviour of the mask-wearing user group. Nine additional participants were instructed to approach the mask-wearing group, directing them to move back until reaching the end of the stage. The two collectives both march between the right and left extremities of the stage, constantly maintaining a mutual gaze and a forward distance of approximately 6 to 10 feet (1.8 to 3.0 m) from the closest person opposite (Figure 3(iii)).

### 3.6. Study Outline

Our study derives from the experimental framework of Performance-Led Research in the Wild [124]. This compromise relieves some of the social confines imposed in a lab setting while bypassing the highly unpredictable nature that comes with arbitrary participation [125]. In this current research action, we set up an open call, as a result of which 12 local artists were chosen to take part in a 2-week residency that would conclude with a public performance. This was structured to build upon compositional practices that were stimulated by a series of performance and improvisation workshops that took place 3 weeks prior. From the residency group, 4 participants were selected to use the wearable during the final performance, presenting the three scenes described above, which lasted just over 8 minutes in total.

We recorded video and sensor data throughout each scene in order to evaluate the qualities of interaction. From this sub-group, 3 identified as male and 1 female with ages between 24 and 51 years old. All 4 held long-term experience as highly-skilled musicians, performing internationally and retaining regular musical practice. An additional 4 participants, 50:50 split male and female, trialled primitive iterations of the system during

the prior rehearsal stages, from which we produced field notes and recorded user feedback. Aside from a handful of local encounters, the user group was considered strangers amongst one another, not closely bonded and only acquainted during the preparation stages.

The first study took place in Lisbon, Portugal between October and December 2020, before the national vaccination rollouts were initiated. Cultural activities, such as live performances, were permitted in accordance with the measures set out by local authorities [8]. In the scope of COVID-19 distancing measures, the user groups were respected as a support bubble throughout the research period, assuring that in-person interaction was permitted under the same conditions as those they share a home [126]. With that said, at no point would the sessions explicitly require a disruption to the standardised safety regimes, particularly those concerning close contact and sanitation of shared materials to minimise viral transmission.

## 4. Preliminary Results

Here, we summarise the perspectives of the individual participants and indicate moments where the first-person accounts are complemented by the acceleration data. We apply statistical analysis to the collective sensor pool to discern group-level features, later used to substantiate relevant design guidelines for proxemic intervention strategies. We acknowledge the analytic restraints when subjected to a single recording from an individual user group that by no means should be taken as a comprehensive survey of the proxemic orchestration. This provisional study is purposed to demonstrate novel adaptations for proxemic interaction and prescribe design insights for future work.

### 4.1. Data Collection

Throughout a 2-week prototyping period, we recorded participant observations and first-person experiences, based upon the user-centred design actions proposed by Bernardo et al. [127]. This was comprised of eight day-long rehearsal sessions that each allowed one hour focused on exploring the wearable followed by a brief interview with participants, asking about any limitations that were discovered and features of the interaction that influenced their behaviour. These sessions took place alongside a course of improvisation exercises fitted towards musical performance. After the study, we extract comments from a series of recordings and field notes. Participants were made aware that they were taking part in a study set out to evaluate the general usability of the wearable device and sound feedback mapping.

We recorded video and audio of the final performance and rehearsals using a single-point microphone and camera. The video clips were colour graded to obtain a clearer view, especially when there was a lack of ambient light being projected onto the stage. When probing deeper into a handful of significant events depicted by our written assessment, we extract still frames from the recordings and overlay these with graphical annotations to comment on the positional qualities present. Audio was processed through Sonic Visualiser (Sonic Visualiser: https://www.sonicvisualiser.org/doc/reference/4.4/en/ (accessed on 25 June 2022) for feature extraction. The sensory masks were embedded with an accelerometer sensing component located at the back of the head, recording three axes of directional acceleration from each user. The triaxial sensor data are unified by computing the Signal Vector Magnitude (SVM), representative of the combined acceleration coordinates x, y and z, as validated by Ward et al.'s proposed method for monitoring synchronous motor behaviour within a group of live performers [128]. Each accelerometer was recorded at 10 samples per second, with the data being synchronised and smoothed in post-processing.

When probing deeper into a handful of significant events depicted by our written assessment, we extract still frames from the recordings and overlay these with graphical annotations used to commentate on the positional qualities present. In Figure 4, we present the median average magnitude acceleration measured from the four users to represent the collective movement, indicated by a series of peaks throughout the three scenes. The median acceleration timeline (top) serves as a guide to navigate the general

progression of movement qualities presented throughout the performance, while separating the data stream for each individual user (middle), and assessing the alignment of the peaks, provides a numerical evaluation of interpersonal responsiveness or absence of. Finally, the peak acceleration points are partitioned according to a series of successive bursts over time (bottom), detected using a k-means clustering algorithm. The clusters help us to convey the collective movement dynamics and relationship qualities that are observed throughout each scene, partly revealed in the average duration and amplitude of the clusters.

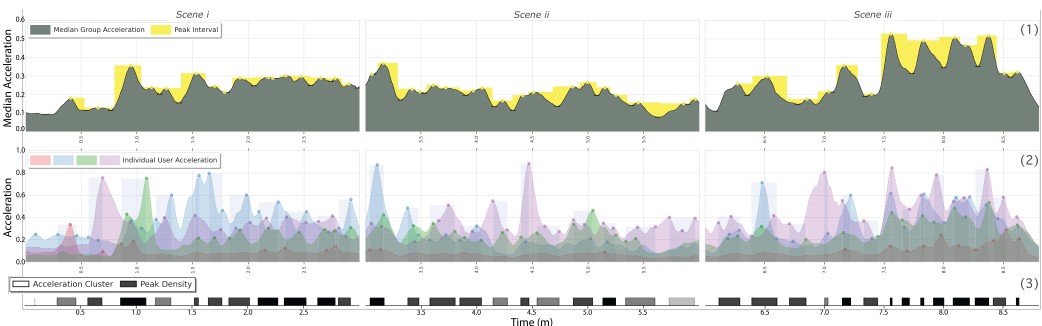

**Figure 4.** Acceleration data recorded from scenes i–iii: the top row displays group median averages (1), with individual user data shown below (2). The final row aligns the peak cluster periods detected along the *x*-axis (3). A high-resolution version of the image is available in the Supplementary Material.

We formulate the following clustering features: Peak Density, the total number of peaks detected relative to the duration of the cluster, and User Effort Dissimilarity, based on an equal alteration of active users.

$$\text{Let } C(x) = \sum_{i=1}^{n} [s_i = x]$$

$$\begin{matrix} \text{User Effort} \\ \text{Dissimilarity} \end{matrix} = \sum_{i=1}^{n} \left( \frac{s_i}{C(x)} - \frac{1}{n} \right) \cdot 100. \tag{1}$$

The User Effort Dissimilarity is calculated according to the proportion of peaks exerted by each individual user, where $C(x)$ represents the total number of acceleration peaks, $s_i$ being those exerted by each individual user.

### 4.2. Interpersonal Synchrony

During Scene iii, we observed participants engaging more confidently before and after the sound mapping is disrupted by the unmasked participants. In these moments, the mask-wearing group march in parallel alignment, maintaining a consistent pace with each other. Additionally, we note the persistent use of eye contact towards the unmasked group, voiding obstruction until reaching the end of the stage. As the procedural soundscape is overridden by the choral chants voiced by the unmasked performers, we see the groups disperse in the opposite direction, subverting the momentum and common alignment. This interplay repeats itself 4 times, where the two sub-groups voluntarily hand over the leading role of the march, alternating every 6–10 s. In Figure 5, four masked performers walk towards and away from a group of those without wearable sensors whilst maintaining a forward distance. For these frames, the annotator was tasked to visually mark the alignment and walking direction of the mask-wearing group.

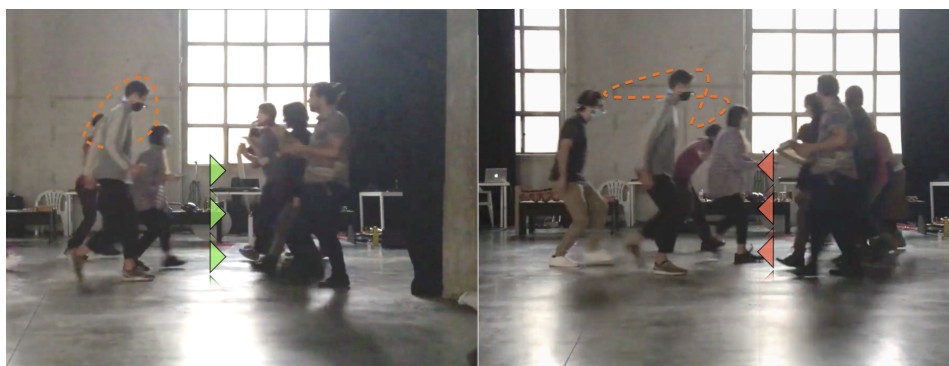

**Figure 5.** Annotations from Scene *iii* rehearsal video. The arrows show the walking direction with a dashed line to trace the dispersion of mask-wearing group. A video recording is included in the Supplementary Material.

We compare the synchronous activity of each scene based on the separation of the clustered peaks. From this measure, we determine that the most consistent levels of synchronous movement take place during Scene iii, with far fewer occurrences during Scene ii. Both scenes have a similar number of peaks per cluster (Table 1), with a greater density of peaks in Scene iii; this is shown by the constrained duration of the clusters formulated in more consistent intervals. This perception is reassured by the increased amplitude of acceleration values throughout, perceived as an intentional syncopation of movements that are highly responsive to one another, and agreeable to the feedback given by one of the users.

*"we had only a short moment to respond, similar to a call-and-response situation you sometimes see in concerts. The motivation was no longer about controlling the timbre (of the sound), but just to assert dramatic impulses before the others take the lead."*

**Table 1.** Peak and cluster statistics from each scene, calculated from the individual user acceleration data that correspond to rows (2) and (3) in Figure 4.

| Scene | Mean Peak Interval (s) | SD Peak Interval (s) | Median Peak Amplitude (g) | |
|---|---|---|---|---|
| i | **183.7** | **149.9** | 0.31 | |
| ii | 182.1 | 141.4 | 0.29 | |
| iii | 155.8 | 138.2 | 0.36 | |
| | User Effort Dissimilarity (%) | Alternating User Peaks (%) | Mean Cluster Concentration (Peaks/Cluster) | Mean Cluster Duration (s) |
| i | 25.5 | 90.2 | **4.5** | 8.9 |
| ii | **44.7** | 78.7 | 3.8 | **10.0** |
| iii | 17.2 | **96.6** | 4.1 | 5.6 |

Our observations and data taken from this final arrangement indicate vastly different group behaviours from before. We would suggest that the major influence lies behind the "call-and-response" routines as the opposing bodies assert themselves into the space, inducing a series of structured interruptions that we probe deeper into in the Discussion, Section 5.2. We will not neglect here that these behaviours were inherently dramatised under theatrical persuasion, but, nonetheless, insist that the core reactions are authentic to the invasive confrontation as the opposing bodies assert themselves into the space.

### 4.3. Spontaneous and Sustained Engagement

Given the premise that regular social engagement is an effective precursor to community wellbeing, generating and sustaining interactions are both considered to play a key role in urban design practices [129–131]. We question how sensor-based mediation can be

used to not only provoke, but actually extend the longevity and quality of a new encounter, described as "what makes the experience comfortable, interesting, and meaningful" [131]. In this case study, we gauge spontaneous engagement from the rotation of users asserting themselves as a leading influence (i.e., a new person exerting an acceleration peak), and how peaks are partitioned between individual users. Sustained interactions are more nuanced, recognised as lingering behaviours that take place amidst the succession of new movements, for which we draw upon the user's control over the sound output.

Responding to our research question RQ2, we enquire into the user's relationship with the external mediation materials, and the way these momentarily interfere with the continuity of the sound output in Scenes ii and iii, separating these external influences by their level of predictability. In Figure 6, we recognise a major disparity in the arrangement of the non-structured interruptions incited by the flying drone compared to those anticipated by the unmasked group (i.e., structured interruptions). The unpredictable manoeuvres imposed by the flying drone disrupt the course of sustained dyadic gestures, from which we dissect the rapid, unplanned interchange of users controlling the sound output. Referring back to Figure 4, this parallels with sporadic changes in peak intervals, with individual exertions coming at arbitrary intervals. Scene iii, on the other hand, the course of sonic deviations better emphasises a uniform sequence of interruptions. This repeatable turn in controllability infers an attentive reciprocation with the other users, showing an advancement from sporadic reflex actions to meaningful group gestures, also granted by steadier peak intervals and the overall progression of synchronous movement that is detailed in the subsection above. One participant describes an impulsive reaction while being approached by the flying drone, suggesting its non-human form carries a provocation for spontaneous engagement:

> "the drone would keep coming closer, like I was being chosen out of the crowd, so I would start following. I was comfortable with running towards the wall or even a flying object, but into somebody else, it's not the same. Even if we do not get so close, I feel like I am threatening someone or just being a nuisance."

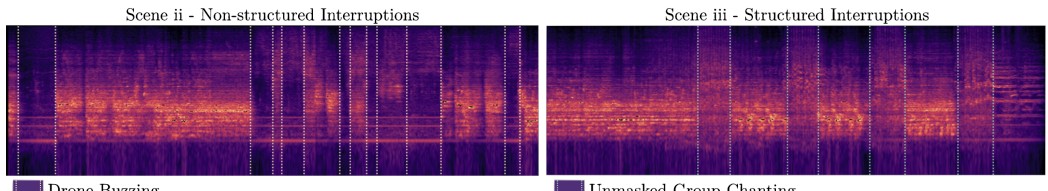

**Figure 6.** Spectrogram recording from two scenes, *ii* and *iii*, marked with interruptions of the granular soundscape.

This correspondence is not so apparent in the acceleration data, where we actually notice a drop in energy regarding the lower mean cluster density and substantial User Effort Dissimilarity, supposing an imbalance of individuals dominating the space. However, we may testify that the shared positional influence of the drone on stage engaged all users in the shared space, even when at a standstill.

### 4.4. Technical Venerability against Proxemic Awareness

In hindsight, the placement of the sensors of the face, constantly shifting their projected angles meant that misreadings were highly probable. When tested in lab conditions, these proximity sensors are expected to perform reliably at differing trajectories [116], but when we inspect the sound recording and video footage, we observed many cases of unexpected readings during the performances in contrast to the testing phase. Participants would move closer together with no perceivable feedback. Moreover, the detectability of the ultrasonic reflections was highly dependent on the material of the occlusion. Table 2 provides a set of sample signals recorded when approaching the sensor using three different surfaces. These were recorded separately in controlled conditions. The ceramic tile provided the greatest

detection range, while the clothing material resulted in more noise and vulnerability to drop-outs. Though our signal filtering methods helped remove extreme anomalies, there would still be a great deal of inaccurate measurements being fed into the system. For these reasons and more, this particular sensor technology has been discouraged for measuring interpersonal distance in a review of wearable devices for proxemic interaction, which exposes multiple scenarios by which ultrasonic sensing was proven to be partly inadequate [107].

Fdili Alaou writes about the perceived messiness that inevitably comes with adopting personal tracking devices into performance-based practice (noise, sensor placement, classification failure) [132]. However, instead of seeing these as issues that need to be resolved, actually encourages artists to embrace the technical nuances, turning technology resistance into creative material. It is important to appreciate that, in non-lab conditions, imperfections will constantly prevail and, therefore, we should welcome and validate the experiences that come with each iteration. In spite of such cases where the sound–movement relationship was not sensible to the performer or even the audience, as revealed repeatedly during the user studies, we contend that there remained a genuine influence from the physical artefact alone. Throughout the study, users demonstrated a strong awareness of their surroundings to control the sounds, recognise technical faults and overcome them through persistent trial and error.

**Table 2.** Comparison of reflected materials from benchmark test recordings. From left to right: ceramic tile, skin from the hand, torso covered by clothing.

| Material | Minimum Detection Range | Maximum Detection Range | Dropout Rate |
|---|---|---|---|
| Ceramic Tile | 5.1 cm | 288.7 cm | 0.0% |
| Skin | 10.2 cm | 246.4 cm | 3.85% |
| Clothing Fabric | 25.4 cm | 201.8 cm | 4.55% |

## 5. Discussion: Preparing for Public Instalment

### 5.1. New Sociable Space

Modern communication technology has been shown to facilitate rich social engagements in a remote setting, by which some degree of face-to-face affairs continue to be viewed redundant, even after confinement measures have subsided [133]. We, therefore, propose that solutions should also exist to support meaningful discourse from an extendable distance that is suitable for the social environment and individuals involved. Taking sound feedback as the core of proxemic mediation, we adopt Mehta's new *sociable space* concept into our design considerations, this being the capability to capture and hold one's attention from a comfortable distance, encouraging spontaneous encounters with non-acquaintances before confrontation into personal space [32]. Inviting flexibility to traditional proxemic theory, we can call upon Roudposhti's behavioural model that was introduced in Section 2.6, demonstrating mannerisms of Indicator and Interest.

While this desirable social dynamic is provisional to a wide spread of design implications, we put forth the benefit of authorising drop-in and drop-out participation, whereby the formation of a group can be altered at any point to openly accommodate new users. Additionally, this functionality insists upon an agreed focal point of interest that establishes a proxemic cornerstone regardless of the group's composition, continually subject to change. We find this inclusive control structure to be very much customary in the context of interactive installations, such that the initiation of the sound feedback invites new unsuspecting members to engage before even being aware of the artefact's existence [134,135]. In our performance-led study, we were enlightened through the inclusion of foreign mediation artefacts, acting as a vital component for exposing new dynamics of the sound, shown less prominently when the mask-wearing group was isolated from the external surroundings in Scene i.

This draws upon the gestural limitations while adhering to a minimum interpersonal distance of 4 feet (1.2 m), absent from any positional cues to prompt collective engagement.

### 5.2. Proxemic Sensibility and Sensitivity

Our literature review in Section 2 supports the vital influence that routine social interactions have on public spaces and the proceeding benefits that come with this. We also bear in mind the negative view that undesired social isolation can be detrimental to one's mental condition [21,22,136]. The pandemic forced prolonged periods of isolation that would ultimately cause a rise in self-reported loneliness [137], reported to be particularly harmful for those who already experienced anxiety prior to the pandemic period [19,138]. Such conditions have been shown to suppress one's tolerance for engagement within intimate proxemic boundaries [139]. We can also include recent findings related to isolation and cognitive function, locating harmful effects on the brain region associated with spatial orientation, learning and memory [20,140], which presumably foreshadows a long-term disassociation with in-person social situations. A critical motivation behind our work is to examine what interventions open up a safe intermediate to re-socialisation for individuals who do not yet feel comfortable exposing themselves in public [141], understanding that each person will hold their own preference for personal space.

Moving away from a standardised proxemic model, we form an empathetic view around personal space, appreciative of boundaries that are unfixed and individualistic in light of one's past experiences and various other factors that are undisclosed between strangers. Reinforcing Pentland's descriptors assigned previously, this derives from the sentiments that embody Empathy and Interest. Addressing RQ1, we articulate the call for proactive awareness as part of the following design considerations with regard to sensory intervention, *Proxemic Sensibility* and *Sensitivity*. Sensibility is the altruistic responsibility that ensures safe coordination of bodies, mindful of the surroundings and presence of any individual, paired with Sensitivity, for those to stay receptive to the actions projected by others, with the willingness to alter their paths accordingly. We still cannot be certain of the system's influence without any sound feedback since the study does not include a specific control condition for this. However, given the mixed experiences that arose in the presence of external interactive artefacts on stage, these being the flying drone and the unmasked participant group, we speculate on the suitable conditions for effective social signalling through the participatory engagement with sound. Rather than trying to incorporate all of the users simultaneously, we suggest that individualistic control mechanisms can help to elevate one's agency to the surroundings, while the anticipation of regular interchange is necessary to preserve attention. Consequently, we favour the use of structured interruptions by way of turn-taking procedures, whereupon users are compelled to listen to one another, then allocated a sufficient time window in order to react accordingly.

### 5.3. Constrained Complexity

This work fits into the domain of proxemic interaction strategies, asserting novelty in the non-categorisation of physical distances. The early phases of experimentation lead us to try complex, nonlinear sound-distance mapping strategies without the anticipation of hindered usability. However, we were enlightened early into prototyping that such sophisticated mapping strategies are only intuitive on the presumption that users are sufficiently acquainted with the apparatus and expected results, reiterating discussions around the virtuosity of musical instruments within the NIME research community [142]. We first insisted on more abstract mappings, layering sound elements each influenced by multiple users simultaneously; this approach was designed to provoke collective actions and, consequentially, for the group to act upon the movement patterns that made the most sense to them. From this composite sound feedback strategy, however, a great deal of confusion emerged from new users, leaving them with a feeling of disempowerment. We found this ambitious setup expected far too much knowledge from new users, made clear during testing, where participants would express their frustrations with the system's

behaviour while not understanding how their individual actions influence what they were hearing. At a minimum, they would want to walk closer or farther from someone and hear an instantaneous reaction. We were able to recover the user's association to the sound feedback when resorting to a linear distance to amplitude mapping according to the smallest distance detected, allocating control to only one individual at a time in favour of being perceived as more receptive. Though, in many cases, the irregular substitutions would cause excessive shifts in volume, disrupting the continuity of the controlling gesture. In particular, when incorporating external objects and additional users onto the stage, we find more instances of abrupt sound bursts, depicted by the intermittent gaps in the audio recording, as shown in Figure 6.

While working in this semi-controlled setting, we benefited from the user's past experiences using digital musical instruments. However, when gearing toward a public space intervention, the importance of bridging with broader audiences would become most salient, to engage those unfamiliar with the system and each other, as described in Section 6. Such a pursuit calls for an accessible means of individual control that also encourages collaboration amongst strangers. Here, we advocate for a turn-taking framework that operates on a fixed time interval. This conveniently aligns with the step sequencer metaphor proposed in Bengler and Bryan-Kinns's work with non-musicians, studying how a constrained control paradigm can improve engagement with the general public [143]. To build upon these considerations, we prescribe the function of sequencing for distributed control and attention in group interaction, maintaining the usability of single-user mappings whilst emphasising the quality of observing the other.

*5.4. Precautions for Public Inclusion*

The change in pandemic circumstances that happened during the research timeline meant that only a maximum of four participants were authorised to use the wearable at any one time, dismissing any substitution of users between daily intervals. This compromised the conditions for open public inclusion while the measures gradually became stricter, bringing serious doubts in justifying any sort of artistic action that persisted in congregating different social bubbles. In these circumstances, we were confronted with the fragility of the sensory components embedded onto the mask. More often than not, assistance was required from the workshop coordinator to secure the wearable around the user's head; this would unfavourably call for physical contact near the mouth, posing an additional risk of viral transmission. That considered, a misfortune such as this would have been far more problematic if widespread into public hands.

The deferred extremity of the situation was only realised around one month later, as these measures would come forth as a final phase of provisionary measures before the nation was required to fall under a compulsory confinement period (i.e., lockdown). On these terms, the study was committed to a public space intervention with elevated awareness of safety and robustness, particularly in regard to the exchange of wearable components between alternating user groups.

At the time of reorganising the study, this absence of open public participation was deemed somewhat a pitiful solution, albeit one that avoids abandoning the in-person field study indefinitely. Though, in hindsight, we realise these intermediate steps were absolutely necessary before the system could be made freely accessible to the public, ensuring usability and safety to a minimum standard. This mentality of turning restrictions into opportunities for precaution is highly acclaimed in the discussions drawn out from Howell et al. [49]. This work rationalises the function of private space experimentation which would merit design guidelines for public interventions, detailing concerns around safety that would omit the likelihood of welcoming fruitful interactions between strangers. Nonetheless, due to a lack of public exposure, the opportunity arises to grasp deeper insights from the user group. We factor this progression into our design considerations, insisting that new systems intended for public use should first be evaluated in a low-risk environment for a substantial period of time, taking opportunities to carry out data collection and open-ended experimen-

tation. The controllable nature of the performance-led study allowed us to gain insights from a consenting user group, willing to follow instructions, undergo trial & error testing, and contribute to data collection. As a result, we were able to construct benchmarks for usability, later informing design adaptations for pubic inclusion.

## 6. The Case for Public Interventions during a Pandemic

Taking what was learned during the first phase of experimentation and preliminary results, we introduce an adaptation for a proxemic-based sound intervention designed for open urban spaces, which materialised some months later. The installation was publicly active during October 2021; at this point, the fully vaccinated population was reported at over 80% [144], and the majority of cultural actives could resume, as the most prominent pandemic regulations were already lifted [145]. This demonstration was set out to provoke collaborative engagement through sound, similar to what was observed during the closed user study, but in contrast, situated in the public space format. The system was installed for three full days in a touristic square close to the city centre, completely open to any passing members of the general public. Participants were expected to make use of the system independently, and offered only a set of essential instructions that were made accessible online through a mobile web application. Enlightened by our design considerations, we detail an alternative method for distance-to-sound interaction and assess to what extent, this intervention was capable of preserving the qualities of interaction that were preconceived in the discussion. Here, we evaluate the following strategies: drop-in and out participation, structured interruptions, and sequencing.

### 6.1. Instrumentation and Physical Arrangement

The public sensing environment brought many challenges to the system's physical orchestration. The major conditions here may be subject to the installation sustaining itself outdoors, and the considerations in order for it to operate independently without facilitators. We were also intrigued to experiment more deeply into proxemic affordances shaped by the surrounding environment and non-human objects. First off, we reconsidered the mask-worn device to grant user independence, improve safety against viral transmission and refrain from dealing with inaccuracies. As an alternative, four proximity sensors were secured around the lower branches of a tree, pointing slightly downward to establish a path from the tree's crown to a seating area set up 16 feet (4.8 m) away from the base of the tree.

Along with each sensor hangs a brightly coloured ribbon from the branch, representing the origin of the individual paths, indicating the course for users to walk under, as annotated in Figure 7. The extent of the projected sensing area was bounded by the maximum sensing distance, capped at 12 feet (3.6 m) to maintain reliability. Granted that the sensor's performance is dependable on the ambient temperature and humidity, we noticed the detectable range varied more throughout the day than when we were working indoors. The air temperatures would fluctuate from 9–20 °C daily, with generally worsening detection rates during the night.

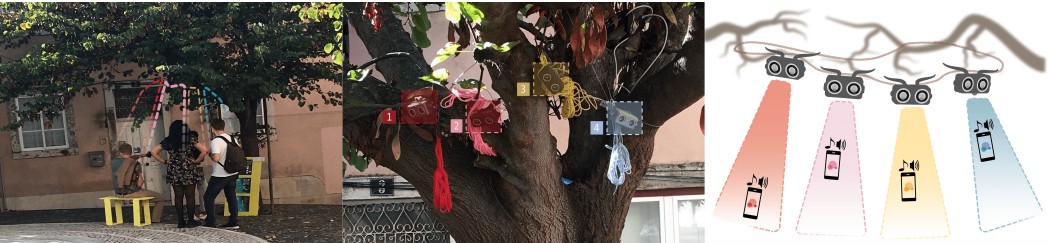

**Figure 7.** Public installation using four proximity sensors placed inside foliage with hanging ribbon. Sensors are physically separated by sensing trajectories, identified by colour.

### 6.2. Spatially Distributed Sound Output

It was in our prime interest that the intervention would preserve the inherent tranquillity of the pedestrian area. To avoid any unnecessary disruption, we insisted that the installation would not exert any sound until participants were willing to engage with the system. We speculated upon a feasible solution by which the public could initiate the sensing mechanism and listen to the installation from their mobile phone. A web app was developed using the Soundworks web framework by Matuszewski et al. [146]. Upon loading the app, participants are prompted to select one of four colours each allocated to one of the sensor placements, determinant of a walking path (Figure 8). The individual sensor measurements are broadcast to each of the mobile phones connected to the current session, activating new notes when someone is detected in the space and continues to do so while the participant moves within the measurable path.

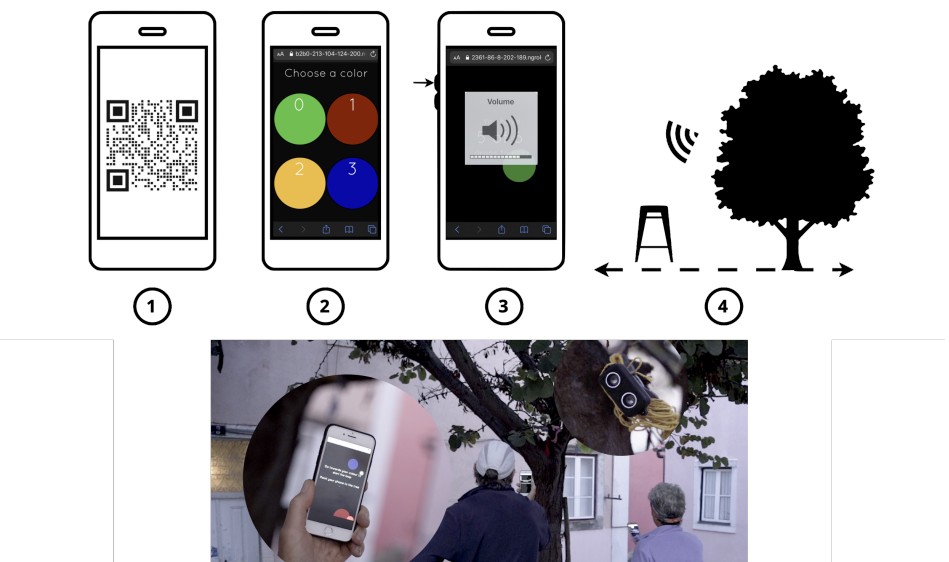

**Figure 8.** User instructions for installation: (1) Access QR code; (2) Load the app and select a colour; (3) Increase the device volume, and (4) Walk towards the chosen colour to initiate sound.

The occlusion distance from each sensor was recorded in consecutive order, cycling through the complete batch every quarter of a second. The distances are converted into MIDI notes on a single octave chromatic scale, allowing 12 possibilities spread out between 30 cm intervals, rising in pitch as the distance measured from the sensor increases. When a user is detected, the app arpeggiates through the incoming notes every cycle. As one note is released, a new note is triggered from the neighbouring user according to their detected distance. Each note is emitted from the mobile device that is assigned to the coloured sensing path, accumulating into short loops that are continuously recorded and echoed in ongoing circulation. A video extract is provided in the Supplementary Material.

### 6.3. From Lab to City, Transferable Experiential Qualities

Over the course of the installation period, we made notes of third-person observational accounts, similar to the rehearsal sessions and performance described prior to this. In particular, we give close attention to moments of group engagement that resonated with the spatial qualities examined previously, thus disclosing the experiences that were transferable from the wearable device to the public space adaptation. This numerical evaluation here does not intend to go as in-depth as the initial study; instead, this closing segment can be considered as a technical primer to fortify our design considerations for future research set in public space environments.

Advancing upon Proxemic Sensibility and turn-taking qualities, we found the mobile distributed sequencer fostered a certain degree of mutual agency, in that participants

would invite their peers to join them and instinctively feel inclined to listen to one another. While exploring different spatial configurations, users would continuously experience new melodic loops with each note coming from separate directions according to the user's position. The separation of the mobile speakers in itself provided an additional modality to the acoustic quality of the soundscape, formed by the relationship between perceived intensity and distance. For the most part, however, users were staring directly at their mobile phones during the interaction process, with their bodies constantly turned towards the sensing apparatus standing parallel to each other, majorly discouraging prospects for mutual acknowledgement through eye contact.

From an observational standpoint, it was not possible to discern synchronous movement patterns inspired by the installation, or any definitive collective movement traits for that matter. Compared to the first case study, resourced with a large stage, the spatial exploration was far more limited here, restricted only to a linear sensing area, revolving around one focal point. Users would insist on staying idle, waiting for the sound to loop for a while, perhaps experimenting with walking forwards or backwards a few steps to trigger new notes. This coincides with the outcomes that presented themselves in Scene ii (Section 3.5) in the way that, when the directional influence of the sensors is displaced from the body and onto the surroundings, users show more infatuation with their own movements over anyone else's. With the ambition to cater for a new *sociable space*, the non-wearable arrangement combined with the drop-in and out functionality of the app was assumed to incentivise a flexible interchange of users, inclined to welcome those non-acquainted into the space. That said, we did not observe any instances of strangers simultaneously engaging at the same time, only those already affiliated, supposedly conditioned to the tensions when asserting oneself into a predetermined social clique.

In this instance, we wish to examine how well the intervention harmonises with the everyday operation of the space, blending with naturally occurring social exchanges, and staying respectful to those not actively participating. We found from a sample of 10 individuals and groups passing by during a weekday afternoon that 7 of 10 would continue walking, three would feel captivated to read the information board with one going as far as entering themselves into the application, and starting to engage with the sound feedback. During the weekend period, the installation would be stationed along the route of a few public walking tours, serving as an amusing artefact to intrigued bystanders, without pulling enough attention for anyone to abandon their personal schedule. In the evening, we observed a spontaneous social gathering take place in the square comprised of 15 or more people mingling beside the installation area. Small groups would approach the installation and briefly engage in a new session, triggering just a few notes before returning back to rejoin the social event. Accepting the severe limitations in retaining interest from new users, these impulsive engagements showcase a strong starting point for public inclusion, by which the artefact successfully captures the attention of broader audiences enough for voluntary initiation. This can partly be owed to our preparation stages, pleading for *Constrained Complexity* to minimise the learning curve.

### 6.4. Limitations of Public Space Adaptations

In this supplementary case study, we were granted the opportunity to confront the challenges of proxemic sensing in public space; this exposed a number of external factors that were less problematic in semi-controlled conditions. To confront our final research question RQ3, we discuss design decisions that were influenced by environmental changes, technical durability and usability, contributing to pervasiveness and inclusion. The non-wearable solution was less susceptible to errors, but at the expense of a confined sensing area. The linear note-based interaction combined with the fixed placement of the sensor improved the system's usability when it was made openly accessible to a public audience. With that said, we believe this approach minimised the user's resilience to unexpected outcomes, and made apparent the moment that sound would stop playing, user's would immediately lose interest and move on to proceed with the rest of their day. In lack

of a firm recommendation here, these outcomes continue to linger onto the feasibility of engaging unskilled users with novel interactive music systems [147], only to be exaggerated in a pervasive setting where prolonged participation is nonobligatory. In its wearable form, the original orchestration was purposed specifically to capture group dynamics by way of geometric and temporal relationships. However, here, we discern that the users were not highly aware of the movements happening around them. This persuasion may be accredited to the animations that are triggered concurrently within the application window, for which the extra stimulation has an overriding effect on mutual engagement, as is inferred by Bryan-Kinns's study [120]. Coinciding with the issues we faced when introducing additional visual elements, the authors ultimately caution against an excessive exposure to non-essential information in order to maintain attention in a collaborative interaction setting.

## 7. Conclusions

This article reflects upon the common attitudes associated with interpersonal distancing and social connection during the most critical moments of pandemia. We speculate upon a spatially-informed intervention to be deployed as part of a performance; this incentivised the design of a sensory face mask, coupled with a system for sound interaction. This wearable orchestration was trailed over a series of workshop sessions, inviting participatory feedback used to refine the physical design and mapping strategies in anticipation to be presented in a live performance setting.

Reflecting upon observational notes and data analysis, we construct design considerations that respond to the pivotal challenges surrounding safe, inclusive re-socialisation in public and in theory, what spatially sensitive systems can offer to overcome such issues. This ultimately calls for an individualistic understanding of proxemic boundaries, giving agency to neighbouring bodies through sequential control, adaptable participation and constrained complexity strategies. We frame our findings in a broader perspective on sensory interventions that are not solely relevant to pandemic measures, generating critical reflections that later inform future developments as we appropriate the system towards urban sensing environments, proving transferable qualities amidst persisting limitations when subjected to the general public.

We outline the standout progressions between the two sensing mechanisms, one situated directly onto the body, the other installed into the surroundings, proving transferable qualities amidst persisting limitations when subjected to the general public. For future work, we foresee the benefit of incorporating a hybrid system comprised of wearable and environmental sensors, suitable for large open spaces in the confidence of robust operation. From here, we also look towards long-term studies with diverse user groups, crucial in forming generalisable conclusions of proxemic behaviour and sound feedback strategies.

**Supplementary Materials:** The following supporting information can be downloaded at: https://www.mdpi.com/article/10.3390/electronics11142151/s1, Figure 4: Acceleration Data Graph; Video S1: Video Extract; Video S2: Performance Video Extract; Video S3: Sound Installation Video.

**Author Contributions:** H.P.D.S. and H.G. are founders of the company responsible for the design and manufacturing of some of the sensory technology used for the wearable presented in the article, specifically the microcontroller component (PLUX Wireless Biosignals). W.P. was an employed researcher with the company during the time of the study. All authors have read and agreed to the published version of the manuscript.

**Funding:** This work has been supported by Marie Skłodowska Curie Actions ITN AffecTech (ERC H2020 Project 1059 ID: 722022).

**Informed Consent Statement:** The research meets all ethical guidelines, including adherence to the legal requirements of the study country. Informed consent was obtained from all subjects involved in the study to publish this paper.



**Data Availability Statement:** All available data will be furnished upon request to the corresponding author, W.P., upon reasonable request.

**Acknowledgments:** We thank the performers for partaking in the study and the audience members for their attendance. Additionally, we acknowledge the work of Bruno Reis for conducting the artistic residency and final performance hosted by Carpintarias de São Lázaro, Câmara Municpal de Lisboa for supporting the public installation. In addition, finally, we thank André Pinto for operating the drone.

**Conflicts of Interest:** H.P.D.S. and H.G. are founders of the company who are responsible for the design and manufacturing of some of the sensory technology that was used for the wearable presented in the article, specifically the microcontroller component (PLUX Wireless Biosignals). W.P. was an employed researcher with the company during the time of the first study.

## Appendix A

1. Adapted firmware and circuitry schematic: https://gitlab.com/wprimett/bitalino-riot-hc-sr04/-/tree/master; accessed on 25 June 2022
2. Adjustable mask strap and microcontroller enclosure:
3. Sensor housing and nasal attachment (back panel):
4. Sensor housing and nasal attachment (front panel):

## Appendix B

1. R-IoT Wireless Latency Tests: in Supplementary Materials.

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
