# Peer review of "Sound Feedback for Social Distance: The Case for Public Interventions during a Pandemic"

_electronics, doi:10.3390/electronics11142151_

Round 1

Reviewer 1 Report

The study evaluates an intervention using wearable sensors during a choreographic live performance. Sensors were attached to the performers’ face masks and were used to manipulate soundscape according to interpersonal distance between performers. Acceleration within the live performance was analyzed. The authors discuss the results with respect to the coordination of interpersonal, social behavior. In addition, the authors discuss experiences with a sound-based distance application in public space

Using a sound-based intervention with respect to interpersonal distance is interesting. Unfortunately, there are major limitations that prevent one to make conclusions about the effects of such intervention. See my comments below:

1) The study does not have any control condition, for example a measurement without sound or with a modified version of the sensors. Due to this lack of control, it is not possible to tell whether the observed effects are in any way related to the intervention.

2) Research question: The research question are not explicitly stated. It is not clear, whether the study was specifically conducted to investigate interpersonal distance during the Covid-19 pandemic or whether another question was studied. What were the hypotheses?

3) Design: A theater performance is a highly artificial setting. Why should results generalize to real-life? I think this needs to be motivated. In addition, placing the sensors directly in the face made them highly salient, this could have major effects on behavior, i.e. when people know that they are being monitored. Also, the face mask itself might have an influence (recently face mask effects on interpersonal distance have been demonstrated in the literature). 

3) There are several open question with respect to the methodology:

- Were any instructions given to the performers? If yes, which? Did participants know about the purpose of the study

- What were the starting positions of the performer for each scenario? What were the distance between performers? Was this controlled?

- How many external users participated in scenario 3? How were external users instructed?

- What kind of interviews were conducted (none of these data are reported in the results)?

- What was the technical set-up? Size of the stage? Number of loudspeakers?

- Please elaborate on the soundscape: Were changes based on distance continuous? The spectrogram makes it look, like there were only two stages. Also, how fast was soundscape modulated? What was the latency between changes in distance and changes in sound?

- Please describe the technical set-up by which acceleration was measured and analyzed

- At which date was the study conducted and where? What were the exact Covid-19 regulations/incidences during this time? Were performers vaccinated?

- In my understanding performers spent a lot of time together during the preparation, therefore I think it is difficult to label them as strangers

- Sensors: For which angles (azimuth and elevation) were distances measured by the sensors? The authors mention that the system failed several times in the live performance. How often was this the case?

- It is not clear how the clustering algorithm worked

- I do not think it is valid to take synchrony in acceleration as a “reliable indicator of prosocial affiliation”. These two might be related but I do not think that any conclusion can be made with respect to affiliation.

- It is not clear how metrics like “user effort dissimilarity” were calculated

- It is not clear how annotation of events based on the videos were conducted. What were the categories? How many persons were annotating the data

4) Results

- The authors do not report any data on distance. Wouldn’t this be the most important metric?

- Given the variance in the data I do not think that any reliable conclusions can be drawn from an 8 minute segment of 4 participants. 

- Why is the x-axis in Figure 7 in seconds? It is not clear what exactly the figure shows

Minor

- Please state distance also in meter/centimeter, not only feet. 

- There are several typos. Overall, language and logical structure it is quite hard to follow. I strongly suggest proof-reading

Reviewer 2 Report

Thank you for the opportunity to review this work. In the spirit of full transparency, my expertise is in the field of communication sciences and disorders. Therefore, I am not as familiar with the work that was cited in this manuscript. In addition, I did not feel qualified to comment on the details of the methodology or innovation.

Overall, I found the manuscript to be difficult to access as a non-expert in this field. The introduction and background were written with the assumption that the reader is intimately familiar with specific concepts and terminology. And, by the end of the introduction, I still wasn't exactly sure what a wearable sensing medium was, why it was necessary, what it would accomplish. My understanding is that the authors want to design a proximity sensory. If this is true, how does this addresses an unmet need?

Given this, it would be helpful to have some clear examples for some of the concepts that the authors are trying to convent. For example:

It is not clear what wearable sensor technology is to a lay person. Do you mean wearable microphones or something else?

Can you write this a bit more clearly (lines 34-35)? "We foresaw an experimental convergence of the spatial qualities of the onstage performers and the tensions on interpersonal conduct carried out in public."

What does the sentence on lines 36-37 mean and how is it relevant to the study? "The notion of physical distance between bodies discloses a high dimension of non-verbal social cues"

In addition, the background seems to include various topics, but it is difficult to see the relation among them. For example, there is discussion about performers as well as design of urban spaces. Later in the discussion, it is clear why these issues were discussed in the background. But, that relation should be made clearer in the background. My suggestion is to simplify the language and write more clearly for non-experts.

Line 332: Why can't the wearer of the device use their own vision to gauge the presence of another person?

Figure 4: The legend in the figure is very difficult to read.

It seems like the case study of the proximity sensor in the public space is better suited for the results than the discussion.

Round 2

Reviewer 1 Report

I appreciate the authors thorough responses and I have only two comments to add:

1) Thank you for pointing out the research questions. I agree that the manipulation of “foreign mediation materials” (I assume that this refers to interacting with humans vs non-human?)can tell us something about in which contexts sound feedback influences behavior. However, I still think that this does not eliminate the need for a control condition, where the same interactive context is testes with the no or an altered version of the sound scape. As a consequence, I do not understand how RQ1 (How can sound feedback improve collective awareness and sensibility to interpersonal distance ) can be answered. This should be listed as a limitation.

2) Please add information about the latencies of the system (i.e. how long did it take from registering a change in distance to presenting an altered version of the soundscape). In my view an immediate response of the system is crucial for the successful implementation of such a system and therefore for the current study.

Author Response

We thank the reviewer for this feedback in response to the first rebuttal. We would like to propose the following changes to address the concerns that remain. Please see our summary below. 

1) We should mention also that RQ1 is only brought up in the discussion and not the results, and is used mainly as preparation for Section 6. With that said, we appreciate this comment and we will open up to the limitation when addressing RQ1:

We still cannot be certain of the system's influence without any sound feedback since the study does not include a specific control condition for this. (line 616)

also, to clarify:

Giving attention to RQ2, the following passage redirects the proxemic attention from the neighbouring bodies, onto the surroundings... (line 350)

2) We include the following details regarding latencies:

At this sample rate, the maximum response time is accepted as 100 milliseconds plus any wireless latency, averaging at ~11 milliseconds in such conditions. (line 276)

We have previously published a report of the expected latencies with these devices that we will refer to here, and ideally to be included in the Supplementary Materials.

Appendix B. Technical Specification

Appendix B.1 R-IoT Wireless Latency Tests - R-IoT Wireless Latency Tests

Round 3

Reviewer 1 Report

All my points have been adressed.